# Comparison of CLEIA and ELISA for SARS-CoV-2 Virus Antibodies after First and Second Dose Vaccinations with the BNT162b2 mRNA Vaccine

**DOI:** 10.3390/vaccines10040487

**Published:** 2022-03-22

**Authors:** Mohammad Said Ashenagar, Akiko Matsumoto, Hironori Sakai, Mikiko Tokiya, Megumi Hara, Yoshio Hirota

**Affiliations:** 1Department of Social and Environmental Medicine, Faculty of Medicine, Saga University, 5-1-1 Nabeshima, Saga 849-8501, Japan; sx4932@cc.saga-u.ac.jp; 2Cellspect Co., Ltd., 2-4-23 Kitaiioka, Morioka 020-0857, Japan; hsakai@cellspect.com; 3Department of Preventive Medicine, Faculty of Medicine, Saga University, 5-1-1 Nabeshima, Saga 849-8501, Japan; harameg@cc.saga-u.ac.jp; 4Clinical Epidemiology Research Center, Medical Co., LTA(SOUSEIKAI), Higashi-ku, Fukuoka 813-0017, Japan; hiro8yoshi@lta-med.com

**Keywords:** antibody, COVID-19, SARS-CoV-2, immunoassay tests, vaccination

## Abstract

The global severe acute respiratory syndrome coronavirus 2 (SARS-CoV-2) pandemic has required rapid action to control its spread and vaccines are a fundamental solution to this pandemic. The development of rapid and reliable serological tests to monitor the antibody response to coronavirus disease vaccines is necessary for post-vaccination immune responses. Therefore, in this study, anti-SARS-CoV-2 antibody titers after the first and second doses were monitored using two different measurement systems, a highly sensitive analytical platform of chemiluminescent enzyme immunoassay (CLEIA) and an enzyme-linked immunosorbent assay (ELISA). Our study included 121 participants who received two doses of the BNT162b2 vaccine. Both methods show significant increase in anti-spike protein IgG antibody levels one week after the first vaccination, and then reached at a plateau at week five (week two after the second dose), with a 3.8 × 10^3^-fold rise in CLEIA and a 22-fold rise in ELISA. CLEIA and ELISA showed a good correlation in the high titer range, >10 binding antibody unit (BAU)/mL. Both methods detected higher IgG antibody levels in females compared with male participants after the second vaccination, while CLEIA exhibits the sex difference after the first dose. Thus, our study showed better performance of CLEIA over ELISA in sensitivity, especially in the low concentration range, however ELISA was also useful in the high titer range (>10 BAU/mL) corresponding to the level seen several weeks after the first vaccination.

## 1. Introduction

The novel coronavirus disease (COVID-19) was first reported in Wuhan, China, in late December 2019, and it then spread globally. The WHO declared a pandemic in March 2020 [1]. Rapid responses to the severe acute respiratory syndrome coronavirus 2 (SARS-CoV-2) pandemic are required to control its global spread and vaccines are a major solution to this pandemic. Therefore, vaccine development is being fast-tracked globally and has led to new vaccine types (e.g., DNA, RNA, inactivated forms of virus) [2]. More than 10 billion doses of the vaccine have now been administered globally [3]. As of 26 January 2022, 204,528,007 vaccine doses have been administered in Japan [4]. A total of three vaccines have been approved for use in Japan, and the mRNA Pfizer–BioNTech BNT162b2 and TAK-919 (Moderna formulation) vaccines are the most widely used [5].

mRNA-based vaccines avoid the risk of integrating viral genetic material into the host cell’s genome and can produce pure viral proteins. SARS-CoV-2 vaccines are based on virus mRNA, specifically the fragment encoding the spike (S) protein, which attaches the virion to the host cell’s membrane [6]. SARS-CoV-2 encodes four main structural proteins: spike (S), envelope, membrane, and nucleocapsid (N), as well as multiple nonstructural proteins and accessory proteins [7]. After SARS-CoV-2 infection, individuals typically start producing virus-specific antibodies including immunoglobulin (Ig)G, IgM, and IgA, which mainly target two viral proteins, the S protein, and the N protein (NP). The S protein is the outermost protein on the virus surface and contains a receptor-binding domain (RBD) [8,9,10] and for SARS-CoV-2, the S protein is cleaved into S1 and S2 subunits by furin protease before the virion is released. The S1 subunit contains an immunologically RBD, which is a major antibody target [11]. 

Antibody responses may correlate with the level of protection of vaccine-induced immune responses. SARS-CoV-2 mRNA vaccines were proposed to be administered at least twice with a spacing of 21 or 28 days to increase the activation of the immune system. A two-dose regimen of the BNT162b2 mRNA vaccine (Pfizer-BioNTech, NY USA/Mainz, Germany) was found to be safe and 95% effective against COVID-19 [12,13,14]. A recent study indicated that anti-spike IgG levels were associated with protection from infection after two dose BNT162b2 vaccination and even greater after prior infection [15]. However, antibody profiles vary across individuals. The level of S-IgG antibodies was maintained for six months after the second vaccination of BNT162b2 mRNA COVID-19 vaccination among healthcare worker in Korea [16]. In Israel, the individuals who received the two doses of the BNT162b2 vaccine have different kinetics of antibody levels compared to patients who had been infected with the SARS-CoV-2 virus, with higher initial levels but decrease faster in the first group [17]. Another study found that protection against SARS-CoV-2 infection wanes over time after full vaccination with BNT162b2 [18]. Furthermore, limited waning in effectiveness of the BNT162b2 vaccine and a duration of protective immunity was observed in older adults and in those in a clinical risk group [19]. This will be closely monitored in years to come and there will be an increasing demand for reliable rapid serological tests.

Serology-based immunoassays are an inexpensive and rapid testing method for epidemiological surveillance to detect antibodies produced by infected individuals in response to SARS-CoV-2 exposure or vaccine response and predict protective immunity since it has been demonstrated that the COVID-19 vaccines can induce a humoral response thereby protecting individuals from symptomatic COVID-19 [11,20,21]. Effective and reliable serological detection methods have a critical role in monitoring the abundance of antibodies in infected patients and quantifying the quality of immune responses to new vaccines. Several SARS-CoV-2 immunoassays have been developed [22,23]; however, they use different units and have different measurement accuracies, making a direct comparison of measurement values difficult and important uncertainty about the assay accuracy remains. Furthermore, clinical implementation requires validation of these different assays. The highly sensitive chemiluminescence enzyme immunoassay (CLEIA) has gained increasing attention because of its high reproducibility, low cross-reactivity with other coronavirus antigens, and low interference from common blood components [24]. Therefore, this pilot study compared CLEIA and an enzyme-linked immunosorbent assay (ELISA) to detect anti-SARS-CoV-2 antibodies in vaccinated individuals.

## 2. Materials and Methods

### 2.1. Ethics Statement, Participants, and Sample Processing

This study was approved by the Ethics Committee for Clinical Research of the School of Medicine Saga University, Saga, Japan (No R2-44 and R3-9). All participants provided written informed consent before undergoing any study procedure.

#### 2.1.1. Study Design and Participants

The study group consisted of 121 participants (59 hospital staff, 20 healthcare workers, and 42 students) from hospitals in Saga prefecture who were invited to be vaccinated with two doses of BNT162b2 (Pfizer/BioNTech). The participants were aged 22 to 63 years and 43% were male. Some participants (20.7%) reported disease histories of chronic conditions before vaccination (Table 1). The first vaccine dose was scheduled for February, April, and May 2021, and the second dose was administered 21 days after the first dose. We also included a non-vaccinated student from the same university as a member of the students group in this study to observe the antibody production in the same manner used for other participants (negative control).

#### 2.1.2. Serological Tests

Blood samples were collected prior to the first vaccination and every other week after the second vaccination for healthcare workers, four weeks after the second vaccination for hospital staff, and prior to the first dose, three weeks after the first vaccination, and four weeks after the second vaccination for students (Appendix A).

The first dose of the BNT162b2 (Pfizer/BioNTech) vaccine (30 µg dose) was administered at day 0 and the second dose at day 21. Then, immune responses were evaluated by measuring IGs IgM and IgG. Serum was collected on the day of blood collection and stored at −80 °C until analysis. CLEIA and ELISA were used to quantify the in vitro quantification of human IgM and IgG antibodies to SARS-CoV-2. 

### 2.2. Antibody Measurements

#### 2.2.1. Chemiluminescent Enzyme Immunoassay (CLEIA) 

We measured three anti-SARS-CoV-2 antibodies, anti-S1-IgG, anti-S1-IgM, and anti-N-IgG using the high-sensitivity CLEIA platform (HISCL) (Sysmex Co., Kobe, Japan), which was reported by Noda et al. [24]. HISCL is a fully automated immunoassay system using the chemiluminescent sandwich principle. First, the serum sample was placed in contact with SARS-CoV-2-specific recombinant antigens bound to magnetic beads. Then, after a first round of bound/free separation, the antigen-antibody complex was incubated with an alkaline phosphatase-conjugated antibody against human IgG or IgM to form a sandwich immunocomplex. After a second round of bound/free separation, a luminescent substrate was added into the solution to allow for luminescence measurement. Chemiluminescence intensity was obtained within 17 min following substrate addition. In the reaction chamber, the temperature was maintained at 42 °C throughout the procedure [24]. The reproducibility (coefficient of variation, CV values) was previously reported as 1.4% at 3.3 AU/mL and 2.1% at 24.5 AU/mL for anti-nucleocapsid protein IgG, 1.2% at 4.2 Sysmex Unit (SU)/mL and 33.4 SU/mL for anti-S1 IgM, and 3.3% at 4.2 SU/mL and 2.5% at 35.6 SU/mL for anti-S1 IgG. Regarding anti-S1 IgG, SU/mL is converted to Binding Antibody Units (BAU)/mL.

#### 2.2.2. Enzyme-Linked Immunosorbent Assays (ELISA) 

The antibody titers against the SARS-CoV-2 N protein and S protein S1 were measured by a QuaResearch (Cellspect, Iwate, Japan) COVID-19 Human IgM IgG ELISA Kit (Nucleocapsid Protein) (Cellspect: RCOEL961-N) and QuaResearch COVID-19 Human IgM IgG ELISA kit (Spike Protein-S1) (Cellspect: RCOEL961-S1), respectively. 

The antigen immobilized on plates in RCOEL961-N was the SARS-CoV-2 N protein expressed in *Escherichia coli*, and in RCOEL961-S1 it was the S protein S1 expressed in *E. coli*. The N protein (aa 1–419) and S protein S1 (aa 251–660) were derived from the Wuhan strain. Serum samples were diluted with 1% bovin serum albumin (BSA) in phosphate buffered saline with Tween-20 (PBST) for measurement. The dilution ratio was 1:1000 for RCOEL961N, and 1:200 for RCOEL961-S1. For measurements, 100 µL of diluted measurement sample was added to a well on the antigen protein-immobilized plate, and the mixture was incubated at room temperature for 1 h. After incubation, the well was washed five times with 200 µL PBST. Diluted Horseradish Peroxidase (HRP)-conjugated antibody was added, and the mixture was incubated at room temperature for 1 h. After incubation, the well was washed five times with 200 µL PBST and 100 µL TMB was added to each well. Then, 100 µL of 1 M HCl was added to stop the reaction and the absorbance at 450 nm was measured by SH-1000 (CORONA ELECTRIC Co., Ltd., Ibaraki, Japan). The reproducibility of the assays is shown in Appendix A. The coefficient of variation of ten measurements was 5.3–13.0%.

### 2.3. Statistical Analyses

The paired *t*-test was used to compare and evaluate the immune responses after vaccination assessed by ELISA and CLEIA. A mixed model was used to compare log-transformed IgG levels between males and females considering repeated measures and the random effect of the subpopulation (proc mixed). As participants with steroid use did not show apparently low levels of anti-S1 IgG, e.g., ant-S1 IgG in non-users and users showed 3803-fold rise and 3099-fold rise at week five after the first dose, they were also included in the analysis with adjustment (fixed effect). Statistical analyses were performed using SAS9.4 TS Level 1M5 for Windows (SAS Institute, Cary, NC, USA). *p* < 0.05 was considered statistically significant.

## 3. Results

The SARS-CoV-2 N IgG responses assessed by CLEIA and ELISA in this study had no peak after the first and second vaccinations with CV values below 120%, suggesting no participants had COVID-19 infection (Table 2). As a negative control, one of the non-vaccinated individuals was tested four times, at 0, 3, 7, and 15 weeks, and the anti S1 IgG titers were found to be as low as 0.5–1.1 BAU/mL.

As shown in Figure 1, CLEIA showed a significant increase in anti-S1 IgM and IgG levels at weeks two and one after the first vaccination, respectively (*p* < 0.0001 and *p* = 0.0159 by paired *t*-test with log-transformed values, *n* = 19 and 20, respectively). ELISA showed a similar increase in antibody titers (Figure 1). Anti-S1 IgM and IgG levels increased exponentially and reached a plateau at week four (week one after the second dose, a 54-fold increase) and at week five (week two after the second dose, a 3.8 × 10^3^-fold increase), respectively, using CLEIA after the first mRNA vaccination (Figure 1). ELISA showed the anti-S1 IgM and IgG levels peaked at the same time as those assessed by CLEIA; however, ELISA levels were much lower than those determined by CLEIA (1.8-fold and 22-fold increases for IgM and IgG). IgM and IgG showed 7-fold and 1249-fold increases at 15 weeks after the first vaccination (12 weeks after the second vaccination) by CLEIA, and only 1.2-fold and 11-fold increases at week 15 by ELISA (Figure 1).

Figure 2 shows the sex differences in anti-S1 IgG titers; higher IgG levels were observed in samples from females compared with males at weeks one, two, four and five (one and two weeks after each of the two vaccinations) by CLEIA (*p* = 0.007, 0.047, 0.012 and 0.024 at weeks one, two, four and five, respectively); the least square means (LSM) in male and female were 0.7 and 1.8 BAU/mL at week one, 34 and 70 BAU/mL at week two, 670 and 1697 BAU/mL at week four, and 1165 and 2616 BAU/mL at week five. By ELISA, higher IgG levels were detected in females at weeks four–six (one–three weeks after second dose) (*p* = 0.027, *p* = 0.035, and *p* = 0.038 at weeks four, five, and six, respectively); the LSM (in OD450 nm) in male and female was 0.9 and 1.8 at week four, 1.2 and 2.1 at week five, and 1.1 and 2.0 at week six.

Antibody titers measured by CLEIA (upper panel) and ELISA (lower panel) are shown. Data represent least square geometric mean ± geometric standard error estimated by mixed model; interactions between sex and time course were tested in a mixed model considering sex, age, steroid use, and number of weeks (categorical variables) as fixed effects and repeated measures of the same subject and target population (healthcare workers, hospital staff, and students) as random effects. *, *p* < 0.05; #, *p* = 0.056 for interactive effects (sex × time). In total, forty-two students (vaccinations started in May 2021) had blood specimens collected before vaccination and at three, seven, eleven, and fifteen weeks after vaccination. Twenty healthcare workers (vaccinations started in April) had blood specimens collected at all time points. Fifty-nine hospital staff (vaccinations started in February) had blood specimens collected seven weeks after the first vaccination. Arrows indicate vaccination timing. One male student lacked measurements after 11 weeks. One female healthcare worker lacked measurements at week two and after four weeks. Another female healthcare worker lacked a measurement at week four.

As shown in Figure 3 right panel, CLEIA and ELISA testing had a good correlation with the detection of antibodies in a high concentration range (>10 BAU/mL by CLEIA), with a correlation coefficient (r) of 0.9 with log-transformed IgG value; however, no association was observed at a low concentration range corresponding to one–two weeks after the first vaccination (<10 BAU/mL, r = 0.3 with log-transformed values). There was a poor correlation for IgM between CLEIA and ELISA (r = 0.3 using log-transformed values) (left panel). 

## 4. Discussion

This study compared the sensitivity of two immunoassays, CLEIA and ELISA, to measure SARS-CoV-2 antibodies in the blood of individuals vaccinated with BNT162b2 mRNA. In our study group, we showed that ELISA and CLEIA detected increased titers of antibodies consistent with the time-dependent nature of antibody responses to the BNT162b2 vaccine. Our observation indicates that anti-S protein S1 IgG levels significantly increased seven days after the first dose of the BNT162b2 vaccine, which is consistent with a previous report where the onset of protection was observed less than 14 days after the first vaccination in participants without evidence of prior infection [12,24]. The results of this study showed elevated S-IgG levels in participants and remained high after the second dose of the BNT162b2 vaccine, which is consistent with previous studies [15,16]. Another finding in our study was that females had slightly higher immune responses immediately after vaccination compared with males. Previous studies reported that females produced higher levels of S IgG at day 21 [25] and day 28 [26] post-vaccination with BNT162b2. These data suggest that females might develop a stronger antibody response related to differences in hormones that regulate adaptive and innate immune responses compared with males [27].

Anti-S protein S1 IgG antibody levels were significantly increased in the first week after vaccination using the two methods, showing a good correlation between the methods in the high titer range corresponding to the level seen several weeks after the first vaccination, approximately >10 BAU/mL. The CLEIA detected higher IgG antibody levels in females compared with male participants one week after the first vaccination. Therefore, CLEIA and ELISA are useful to monitor antibody levels after full dose vaccinations among a population with normal antibody responses, whereas highly accurate systems may be required in the early stages of immune responses or for people with disturbed immune responses. Antibody testing remains the best method to estimate SARS-CoV-2 infection and positive vaccine responses [28]. However, different antibody tests have different sensitivities. A rapid, appropriate, and sensitive test may be beneficial for serological studies and rational decision-making regarding booster vaccinations. However, a simple method that does not require special equipment such as ELISA is advantageous for its wide use in any location.

Our study has limitations. Firstly, the relatively small sample size potentially limits the generalizability of these results. Secondly, the sample consists of three subpopulations which may introduce demographic variability, although we attempted to minimize the effect from subpopulation by mixed model.

## 5. Conclusions

In conclusion, we demonstrated two immunoassays, ELISA and CLEIA, could be used to quantify the acquisition of immunity to SARS-CoV-2 as a result of vaccination, especially when measuring high concentrations of Ig. It was also suggested that a highly sensitive method, such as CLEIA, is required for monitoring low concentrations of Ig. Because of the lack of data on the persistence of immunity acquired after vaccination, it is important to monitor the level of antibodies over time. Such assays will be an important part of clinical and research studies as well for forming public health policy guidelines.

## Figures and Tables

**Figure 1 vaccines-10-00487-f001:**
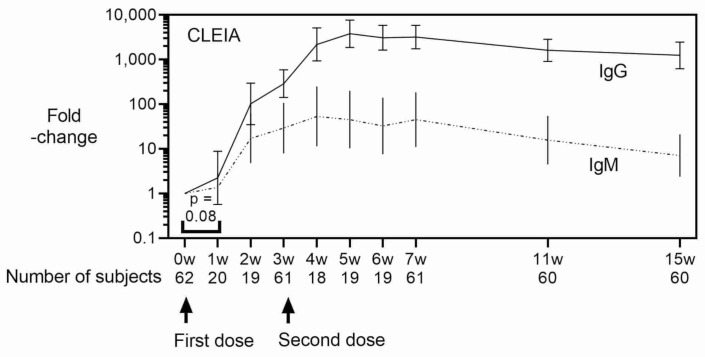
SARS-CoV-2 S1 protein-specific antibody levels measured by CLEIA and ELISA. Antibody titer ratios measured by CLEIA (upper panel) and ELISA (lower panel) are shown (geometric mean ± geometric standard error). Forty-two students (started vaccination in May 2021) had specimens collected before vaccination and at 3, 7, 11, and 15 weeks after vaccination. Twenty healthcare workers (started vaccination in April) had specimens collected at all time points. Arrows indicate vaccination timing. One of the male students lacks measurements at the weeks after 11. One of the female healthcare workers lacks measurements at week 2 and at the weeks after 4. Another female healthcare worker lacks a measurement at week 4. Arrows indicate vaccination. Unless otherwise noted, differences from pre-vaccination are significant by paired t-test using log-transformed value.

**Figure 2 vaccines-10-00487-f002:**
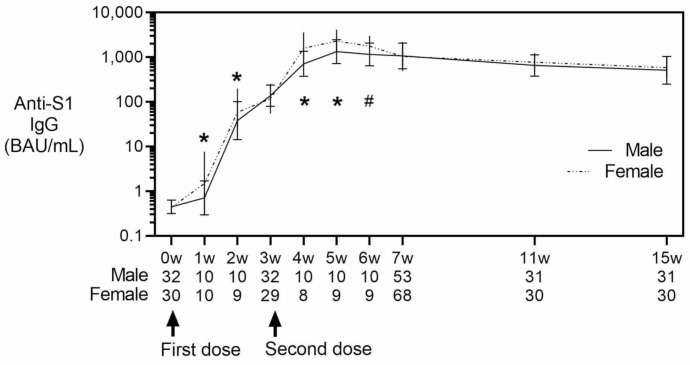
Sex differences in SARS-CoV-2 S1 protein-specific antibody levels. Antibody titers measured by CLEIA (upper panel) and ELISA (lower panel) are shown. Data rep-resent least square geometric mean ŷ geometric standard error estimated by mixed model; interac-tions between sex and time course were tested in a mixed model considering sex, age, steroid use, and number of weeks (categorical variables) as fixed effects and repeated measures of the same subject and target population (healthcare workers, hospital staff, and students) as random effects. *, *p* < 0.05; #, *p* = 0.056 for interactive effects (sex × time). In total, forty-two students (vaccinations started in May 2021) had blood specimens collected before vaccination and at three, seven, elev-en, and fifteen weeks after vaccination. Twenty healthcare workers (vaccinations started in April) had blood specimens collected at all time points. Fifty-nine hospital staff (vaccinations started in February) had blood specimens collected seven weeks after the first vaccination. Arrows indicate vaccination timing. One male student lacked measurements after 11 weeks. One female healthcare worker lacked measurements at week two and after four weeks. Another female healthcare worker lacked a measurement at week four.

**Figure 3 vaccines-10-00487-f003:**
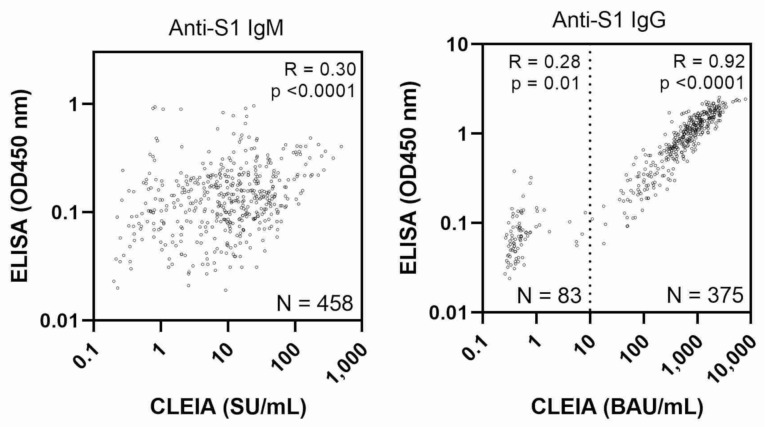
Comparison of SARS-CoV-2 specific antibody levels measured by CLEIA and ELISA. Scatter plots show the levels of anti-spike protein S1 IgG and IgM in serum samples from all participants measured by CLEIA and ELISA. N indicates total number of blood samples taken from participants (Appendix A). The dashed line represents 10 BAU/mL and the plot is divided into high and low concentration ranges. Correlation coefficients (R), *p*-values, and N are shown for each range.

**Table 1 vaccines-10-00487-t001:** Study design and characteristics of participants.

	Healthcare Workers *	Hospital Staff *	Students *
	*n* = 20	*n* = 59	*n* = 42
Study design			
Vaccination			
First dose	April, 2021	February, 2021	May, 2021
Second dose	Three weeks after the first dose, except one participant received the second dose six weeks after the first dose	Three weeks after the first dose	Three weeks after the first dose
Blood samples	Prior to first dose, every other week after vaccination up to three weeks after second vaccination	Four weeks after the second dose	Prior to first dose, three weeks after the first dose, four weeks after the second dose
Characteristics			
Male %	50%	37%	52%
Age			
Median	38	43	22
Interquartile range	30.3–43.5	36.8–51.3	22–23
Disease history			
Sleep disorder	1	1	0
O Hypertension	0	13	1
Diabetes	0	1	0
Dyslipidemia	1	9	0
Kidney disease	0	1	1
Liver disease	0	3	0
Heart disease	0	2	1
COPD	0	5	2
Steroid use	0	4	2
Others	0	4	1

*: Study subjects had no COVID-19 infection before being vaccinated.

**Table 2 vaccines-10-00487-t002:** Level of IgG antibody for SARS-Cov-2 virus nucleocapsid protein.

	Healthcare Workers	Hospital Staff	Students
	*n* = 20	*n* = 59	*n* = 42
Number of measurements	7–8	1	3
CLEIA (SU/mL)			
Median (range)	0.038 (0.01–0.709)	0.037 (0.003–0.228)	0.049 (0.009–0.194)
Intra-individual CV, median (range)	9% (5–30%)	-	11% (0–116%)
ELISA (O.D. 450 nm)			
Median (range)	0.208 (0.063–0.464)	0.09 (0.011–0.626)	0.115 (0.018–0.424)
Intra-individual CV, median (range)	10% (3–31%)	-	11% (1–86%)

## Data Availability

The data presented in this study are available on request from the corresponding author (A.M.). The data are not publicly available due to privacy concerns.

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
