# Peer review of "Comparison of CLEIA and ELISA for SARS-CoV-2 Virus Antibodies after First and Second Dose Vaccinations with the BNT162b2 mRNA Vaccine"

_vaccines, 2022, doi:10.3390/vaccines10040487_

Round 1

Reviewer 1 Report

The authors compared two methods to measure anti-SARS-CoV-2 antibody titers in a time course after first and second doses of the Pfizer/BioNTech vaccine. Comparing different technologies to measure antibodies is important to determine which technologies are better and to aid epidemiologists by presenting accurate assessments of which individuals have sufficient protection against COVID-19.

Major concerns:

1. One of the two major data analysis claims in the Abstract is that:
"anti-spike IgG antibody levels...show a good correlation between the methods in the high titer range"
but the analysis at lines 194-199 and in Figure 3 is unclear and insufficiently detailed.
There are many problems with Figure 3 and its analysis. First, it is unclear how the sample size N=458 was reached since the number of study subjects is 121.
Second, no p-values were computed, so it is unknown whether the correlations are significant.
Third, the text fails to state that the 0.9 correlation is for the high end of anti-S1 IgG (the panel on the right of Figure 3).

2. The following sentence in the Abstract:
"Only the CLEIA detected higher IgG antibody levels in females compared with male par-28 ticipants one week after the first vaccination."
is unclear. What is the adverb "Only" modifying

3. The conclusion that
"Therefore, CLEIA and ELISA are useful for monitoring..."
is too vague. I think what the authors meant is that because the CLEIA and ELISA values are highly correlated in the upper
range for int-S1 IgG, the two tools are reliable in the upper range and other labs can be content to use either CLEIA or ELISA without the
need to use both for validation.

4. A related point to the conclusion being vague is that the Introduction lacks any research questions.
Near the end of the Introduction, the authors should write clearly what question(s) they were trying to answer when they set out to compare the
CLEIA and ELISA antibody measurement tools.

5. In the conclusions, the author write "We showed that the highly sensitive 232 CLEIA method is required for monitoring low concentrations of Ig."
All I understood from Figure 3 is that CLEIA and ELISA are imperfectly correlated. How does it follow from any data the authors showed that
CLEIA is more reliable?

6. In Table 1, it should be stated whether any (and if so, ho many) study subjects had a positive COVID-19 test before being vaccinated.

7. Two weaknesses of the study that should be mentioned in the Discussion are that
-- the sample size of n=121 is small
-- the sample consists of three distinct subpopulations, which may introduce important demographic variability that was not considered in the data analysis.

Minor concerns:

8. The following three statements should be justified by citing one or more published papers:

"After SARS-CoV-2 infection, individuals typically start producing virus-specific antibodies including immunoglobulin (Ig)G, IgM, and 52 IgA, which mainly target two viral proteins, the S protein, and the N protein (NP)."
"Antibody responses may correlate with the level of protection of vaccine-induced immune responses."
"However, the duration of protective immunity is presently unclear, and little is known about antibody responses to SARS-CoV-2 proteins after the second vaccine dose."

I do not agree with the third statement. The following papers (and others) have addressed the duration of protective immunity after vaccination:

Fabiani M, Puopolo M, Morciano C, Spuri M, Spila Alegiani S, Filia A, D'Ancona F, Del Manso M, Riccardo F, Tallon M, Proietti V, Sacco C, Massari M, Da Cas R, Mateo-Urdiales A, Siddu A, Battilomo S, Bella A, Palamara AT, Popoli P, Brusaferro S, Rezza G, Menniti Ippolito F, Pezzotti P; Italian Integrated Surveillance of Covid-19 Study Group and Italian Covid-19 Vaccines Registry group.
Effectiveness of mRNA vaccines and waning of protection against SARS-CoV-2 infection and severe covid-19 during predominant circulation of the delta variant in Italy: retrospective cohort study
BMJ 2022; 376:e069052

Andrews N, Tessier E, Stowe J, Gower C, Kirsebom F, Simmons R, Gallagher E, Thelwall S, Groves N, Dabrera G, Myers R, Campbell CNJ, Amirthalingam G, Edmunds M, Zambon M, Brown K, Hopkins S, Chand M, Ladhani SN, Ramsay M, Lopez Bernal J.
Duration of protection against mild and severe disease by Covid-19 vaccines. New England Journal of Medicine 2022; 386:340-350.

Puranik A, Lenehan PJ, Silvert E, Niesen MJM, Corchado-Garcia J, O'Horo JC, Virk A, Swift MD, Gordon JE, Speicher LL, Geyer HL, Kremers W, Halamka J, Badley AD, Venkatakrishnan AJ, Soundararajan V.
Comparative effectiveness of mRNA-1273 and BNT162b2 against symptomatic SARS-CoV-2 infection. Med 2022; 3:28-41.

The following papers (and others) have addressed antibody response after the second vaccine dose.

Israel A, Shenhar Y, Green I, Merzon E, Golan-Cohen A, Schaffer AA, Ruppin E, Vinker S, Magen E.
Large-scale study of antibody titer decay following BNT162b2 mRNA vaccine or SARS-CoV-2 infection. Vaccines 2021 10:64.

Choi JH, Kim YR, Heo ST, Oh H, Kim M, Lee HR, Yoo JR. Healthcare workers in South Korea maintain a SARS-CoV-2 antibody response six months after receiving a second dose of the BNT162b2 mRNA vaccine
Frontiers in Immunology 2022; 13:827306.

Saure D, O'Ryan M, Torres JP, Zuniga M, Santelices E, Basso LJ.
Dynamic IgG seropositivity after rollout of CoronaVac and BNT162b2 COVID-19 vaccines in Chile: a sentinel surveillance study
Lancet Infectious Diseases 2022; 22:56-63.

9. The word "vaccine" in the title should start with a lower case 'v' or other  words should start with upper case letters

10, In section 2.3, the authors indicate that they used Wilcoxon tests, but I cannot see any results based on Wilcoxon tests

Author Response

Dear Prof. Dr. Tripp,

Editor-in chief

Thank you for giving us the opportunity to submit a revised draft of the manuscript “Comparison of CLEIA and ELISA for SARS-CoV-2 virus anti-bodies after first and second dose vaccinations with the BNT162b2 mRNA vaccine”. We appreciate the time that you and the reviewers dedicated to providing feedback on our manuscript and appreciate for the insightful comments on and valuable improvements to our paper. We have incorporated most of the suggestions made by the reviewers. Please see below, in blue, for a point-by-point response to the reviewers’ comments and concerns. All page numbers/line numbers refer to the revised manuscript file.

Reviewer 1:

Major concerns

  1. One of the two major data analysis claims in the Abstract is that:
    "anti-spike IgG antibody levels...show a good correlation between the methods in the high titer range" but the analysis at lines 194-199 and in Figure 3 is unclear and insufficiently detailed.
    There are many problems with Figure 3 and its analysis. First, it is unclear how the sample size N=458 was reached since the number of study subjects is 121.
    Second, no p-values were computed, so it is unknown whether the correlations are significant.
    Third, the text fails to state that the 0.9 correlation is for the high end of anti-S1 IgG (the panel on the right of Figure 3).

--> Thank you for the comments. We have added the suggested content in Figure 3 (Page 7, Lines 246-251). The p-value and correlation data added in Figure 3.  N indicates the number of blood samples taken from the participants and added Table S2 (supplementary).

  1. The following sentence in the Abstract:

"Only the CLEIA detected higher IgG antibody levels in females compared with male par-28 ticipants one week after the first vaccination." is unclear. What is the adverb "Only" modifying

-->Thank you for pointing this out. We edited it (Line 30).

  1. The conclusion that

"Therefore, CLEIA and ELISA are useful for monitoring..."
is too vague. I think what the authors meant is that because the CLEIA and ELISA values are highly correlated in the upper range for int-S1 IgG, the two tools are reliable in the upper range and other labs can be content to use either CLEIA or ELISA without the
need to use both for validation.

-->Thank you for this suggestion. We edited the suggested point on lines 28-32.

  1. A related point to the conclusion being vague is that the Introduction lacks any research questions.

Near the end of the Introduction, the authors should write clearly what question(s) they were trying to answer when they set out to compare the CLEIA and ELISA antibody measurement tools.

-->Thank you for comment. We have added the suggested content to the manuscript on lines 83-85.   

  1.  In the conclusions, the author write "We showed that the highly sensitive CLEIA method is required for monitoring low concentrations of Ig."
    All I understood from Figure 3 is that CLEIA and ELISA are imperfectly correlated. How does it follow from any data the authors showed that CLEIA is more reliable?

-->Thank you for pointing this out. CLEIA demonstrated far stronger signal intensity than ELISA (3803-fold rise vs. 22-fold rise at 5 weeks, respectively) and the robust reproducibility has been shown previously. Our observation that only CLEIA was able to detect sex effects after first vaccination is consistent with the assumption that CLEIA is more reliable in low concentration range than ELISA. We edited description on lines 227-235 to claim our rationale more clearly.

  1. In Table 1, it should be stated whether any (and if so, how many) study subjects had a positive COVID-19 test before being vaccinated.

-->Thank you for this suggestion. The study subjects had no positive test before being vaccinated.  We added the suggested point in Table 1 (Page 3, Line 106).

  1. Two weaknesses of the study that should be mentioned in the Discussion are that
    -- the sample size of n=121 is small

--the sample consists of three distinct subpopulations, which may introduce important demographic variability that was not considered in the data analysis.

-->Thank you for the comment. We mentioned our limitations by the small sample size (Lines 292-295). We tried to minimize the effect from subpopulation by using mixed model in the analysis of sex effect (figure 2). In the revised manuscript, the figures present the adjusted value. We also confirm the same tendency by stratified analysis for healthcare workers.

Minor concerns

  1. The following three statements should be justified by citing one or more published papers:

After SARS-CoV-2 infection, individuals typically start producing virus-specific antibodies including immunoglobulin (Ig)G, IgM, and IgA, which mainly target two viral proteins, the S protein, and the N protein (NP)."

-->Thank you for suggestion. We added more citation to the text (Line 55).

"Antibody responses may correlate with the level of protection of vaccine-induced immune responses."

"However, the duration of protective immunity is presently unclear, and little is known about antibody responses to SARS-CoV-2 proteins after the second vaccine dose."

I do not agree with the third statement. The following papers (and others) have addressed the duration of protective immunity after vaccination:

-->Thank you for suggestion and the references. We have added the suggested points and references to the manuscript (Lines 58-75).

  1. The word "vaccine" in the title should start with a lower case 'v' or other  words should start with upper case letters.

-->Thank you for pointing this out. We used lower case in the title of manuscript (p.1).

  1.  In section 2.3, the authors indicate that they used Wilcoxon tests, but I cannot see any results based on Wilcoxon tests.

-->Thank you for pointing this out. We deleted it.

Reviewer 2 Report

The manuscript by Ashenagar et al. addresses the importance of sensitive serological assays to detect anti-SARS-CoV-2 IgG and IgM in vaccinated subjects. A study group that includes healthcare workers, students and hospital staff was enrolled for serological investigation to detect anti S1-specific antibodies at different time points after both a single and a double BTN162b2 vaccine administration. The authors compared the sensitivity of two common immunological assays, CLEIA and ELISA, on samples collected from enrolled individuals. They reported and emphasized the higher sensitivity of the CLEIA method in comparison to the standard ELISA technology. Moreover, a sex-correlation within the IgG titer after the second vaccine dose was evidenced, suggesting that women develop a better response after a booster dose. Although the manuscript is interesting, well organized and data well presented, I believe that in this form it is paltry and should be implemented by more interesting and relevant data. Moreover, some issues must be addressed. Within this context, my concerns are the following:

1. Lanes 63-64: provide detailed references about the SARS-CoV-2 anti-spike serological survey on vaccination trials.

2. In the introduction (lane 67) you referred to the serology-based method for monitoring protective immunity against the virus. In my opinion, protective immunity could be detected only by neutralization tests which are not properly neither inexpensive nor routinely used. Standard CLEIA or ELISA does not provide no direct evidences of protection but only a quantification of total IgG. Please discuss.

3. Table 1: participants who regularly use steroids should be excluded since they may develop an altered response to vaccine.

4. Better explain which kind of samples you collected and which one was used for CLEIA rather than for ELISA. At lane 100 you mentioned only serum samples but, further on, you used both serum and plasma for CLEIA (lane 109). Were plasma samples tested also by ELISA? Please correct and comment.

5. Did you include serum/plasma samples from healthy, not vaccinated, subjects? A negative control group is required, mostly to exclude a cross-reaction.

6. Lane 109: the mentioned technology is not a sandwich immunoassay but a standard ELISA or CLEIA method. No capture antibody was used in the assay. Please correct.

7. Figure 1 is quite complex, the references to the abscissa axis are many and could create confusion or misinterpretation. Please, try to make it clearer, if possible. Pleas describe in the figure legend what the N values stands for (also in figure 2). Alternatively, you may show only data about the first dose of vaccine. Since, at this step, you would only evaluate the sensitivity between CLEIA and ELISA, this represents the best group as a lower IgG titer would be expected, thus a more robust assay is required for an accurate detection. Data about the second dose might be reported within the text rather than in the figure.

8. For a better comparison, I suggest to use in the ELISA assay (Figure 2) a WHO standard calibrator and express the results as BAU or as any other quantitative value, as provided for CLEIA. OD values are not properly correct, although widely used.

9. In figure 2, are ELISA data normalized to the cut-off control? OD Values low as 0.1, or lower, seems to be raw data which might be due to cross-reaction. What’s the cut-off value for the ELISA kit? It’s not stated anywhere, nor discussed, as well as borderline values. Mostly, ELISA evaluation is based on an index value.

10. Many studies demonstrated that, among BTN162b2 vaccinated subjects, total IgG titers, along with neutralizing antibodies, are not long lasting and a rapid decrease was observed as soon as subjects received the second vaccine dose. Results that you presented in Figure 1 does not match with this trend, how could you explain that? Please comment in the discussion section.

11. In figure 2 upper panel I would avoid not necessary symbols (#) since the figure is quit complicated. This point is not statistically significant, isn’t it? For CLEIA assay, did you used serum, plasma or both kind of samples? Do you have any information about differences in sensitivity due to sample nature? This could be addressed since plasma could have an higher cross-reactivity in most serological tests.

12. In Figure 2, lower panel (ELISA), you detected an increase in IgG titers in women. Unfortunately, this data was not, or at least partially, confirmed by CLEIA. Since it’s well- established that ELISA is less sensitive than CLEIA, how did you explain this discrepancy in results? A discussion would be interesting and necessary.

13. Please, provide fold values for figure 2 and not only the statistical significance of the results.

14. In the discussion section, lane 208, provide more data regarding the immunological response to the BNT162b2 vaccine in comparison to your results.

15. Lane 208: change “while” with another term.

16. Lanes 222-223: do you have any data about CLEIA and ELISA sensitivity among immunocompromised patients? This would be interesting to investigate and include in the manuscript, if possible.

Author Response

Dear Prof. Dr. Tripp,

Editor-in chief

Thank you for giving us the opportunity to submit a revised draft of the manuscript “Comparison of CLEIA and ELISA for SARS-CoV-2 virus anti-bodies after first and second dose vaccinations with the BNT162b2 mRNA vaccine”. We appreciate the time that you and the reviewers dedicated to providing feedback on our manuscript and appreciate for the insightful comments on and valuable improvements to our paper. We have incorporated most of the suggestions made by the reviewers. Please see below, in blue, for a point-by-point response to the reviewers’ comments and concerns. All page numbers/lane numbers refer to the revised manuscript file.

Reviewer 2.

  1. Lanes 63-64: provide detailed references about the SARS-CoV-2 anti-spike serological survey on vaccination trials.

-->Thank you for the suggestion. We added the suggested points to the manuscript. Lanes 58-75.

  1. In the introduction (lane 67) you referred to the serology-based method for monitoring protective immunity against the virus. In my opinion, protective immunity could be detected only by neutralization tests which are not properly neither inexpensive nor routinely used. Standard CLEIA or ELISA does not provide no direct evidences of protection but only a quantification of total IgG. Please discuss.

-->Thank you for the suggestion. We updated the text (line 77-80) and believe that the CLEIA or ELISA can predict protective immunity.

  1. Table 1: participants who regularly use steroids should be excluded since they may develop an altered response to vaccine.

-->Thank you for pointed this out. In this cohort, the user of steroid did not show apparent inhibition of antibody production, for example, ant-S1 IgG in non-user and user showed 3803-fold rise and 3099-fold rise at week 5, respectively. However, we conducted mixed model with adjustment of steroid use, and replaced figure 2 (p. 6-7) with the one with adjustment. We also altered related descriptions accordingly. We further performed the same analysis without steroid uses, and confirmed the results are almost identical.

  1. Better explain which kind of samples you collected and which one was used for CLEIA rather than for ELISA. At lane 100 you mentioned only serum samples but, further on, you used both serum and plasma for CLEIA (lane 109). Were plasma samples tested also by ELISA? Please correct and comment.

-->Thank you for the suggestion. Serum used in both ELISA and CLEIA and edited (lanes 115 and 123).

  1. Did you include serum/plasma samples from healthy, not vaccinated, subjects? A negative control group is required, mostly to exclude a cross-reaction.

-->Thank you for your comment. Non-vaccinated individuals were observed as negative control (Lanes 170-173).

  1. Lane 109: the mentioned technology is not a sandwich immunoassay but a standard ELISA or CLEIA method. No capture antibody was used in the assay. Please correct.

-->Thank you pointed this out. We used the HISCL method of Noda et al. (2021) which was operated in a fully automatic manner using the chemiluminescent sandwich principle (Noda et al. 2021).

  1. Figure 1 is quite complex, the references to the abscissa axis are many and could create confusion or misinterpretation. Please, try to make it clearer, if possible. Pleas describe in the figure legend what the N values stands for (also in figure 2). Alternatively, you may show only data about the first dose of vaccine. Since, at this step, you would only evaluate the sensitivity between CLEIA and ELISA, this represents the best group as a lower IgG titer would be expected, thus a more robust assay is required for an accurate detection. Data about the second dose might be reported within the text rather than in the figure.

-->Thank you for suggestion. We updated the Figure 1. N indicates total blood samples taken from the participants and included in the legend of Figure 2. The suggested point included in the manuscript (Lanes 215-226).

  1. For a better comparison, I suggest to use in the ELISA assay (Figure 2) a WHO standard calibrator and express the results as BAU or as any other.

-->Thank you for the comment. As the reviewer commented, the BAU is supposed to be calculated using the WHO standard calibrator for a better comparison. Unfortunately, it is currently unavailable in Japan, therefore, direct comparison of measurements by different measurement methods/ institutions is not possible. However, we believe that our analysis is still meaningful, since the statistical analysis is designed to detect inter-/intra-individual variation within this cohort, where comparisons are made by adjusted value (regardless of the absolute value).

  1. In Figure 2, are ELISA data normalized to the cut-off control? OD Values low as 0.1, or lower, seems to be raw data which might be due to cross-reaction. What’s the cut-off value for the ELISA kit? It’s not stated anywhere, nor discussed, as well as borderline values. Mostly, ELISA evaluation is based on an index value.

-->Thank you for the suggestion and question. The results of repeat measurements of low-concentration samples performed in our study, to confirm the accuracy of ELISA, were approximately 0.05 O.D., and the coefficient of variation (CV) of 10 repeat measurements was 5-6% (Table S1). Therefore, the lower limit of quantification, at which the CV of 10 repeat measurements generally exceeds 10%, is considered to be below 0.05 for our ELISA methods. The values measured in this study ranged from 0.024-2.544 for S1 IgG (19 of 458 samples were below 0.05) and from 0.019-0.961 for S1 IgM (45 of 458 samples below 0.05), therefore, we believe that concern on the measurement accuracy of our ELISA is minimal, especially for S1 IgG measurement.

  1. Many studies demonstrated that, among BTN162b2 vaccinated subjects, total IgG titers, along with neutralizing antibodies, are not long lasting and a rapid decrease was observed as soon as subjects received the second vaccine dose. Results that you presented in Figure 1 does not match with this trend, how could you explain that? Please comment in the discussion section.

-->Thank you for pointed this out. We do not think IgG titers rapidly decrease as soon as people receive the second vaccination. We think it rapidly increase after second dose. Other research groups as well as ours have shown it.

  1. In Figure 2 upper panel I would avoid not necessary symbols (#) since the figure is quit complicated. This point is not statistically significant, isn’t it? For CLEIA assay, did you used serum, plasma or both kind of samples? Do you have any information about differences in sensitivity due to sample nature? This could be addressed since plasma could have an higher cross-reactivity in most serological tests.

-->Thank you for the suggestion.

The # symbol represents borderline significance, p = 0.056, therefore we believe the mark is informative, especially in such situation where we need to compare CLEIA and ELISA.

We only used serum and apologize for any confusion caused by the method description. We deleted “plasma" in method section (2.2.1. CLEIA).

  1. In Figure 2, lower panel (ELISA), you detected an increase in IgG titers in women. Unfortunately, this data was not, or at least partially, confirmed by CLEIA. Since it’s well- established that ELISA is less sensitive than CLEIA, how did you explain this discrepancy in results? A discussion would be interesting and necessary.

-->Thank you for the comment. As can be seen in the Figure 2 (pp.6-7), new version with steroid use adjustment, the effect of sex on CLEIA value shows the same trend as the ELISA. Although p values are just across the statistical significance level, we do not believe that there is a meaningful difference between CLEIA and ELISA.

  1. Please, provide fold values for figure 2 and not only the statistical significance of the results.

-->Thank you for the suggestion. We included fold values for Figure 2 (p.6).

  1. In the discussion section, lane 208, provide more data regarding the immunological response to the BNT162b2 vaccine in comparison to your results.

-->Thank you for the comment. We provided more data/information (Lines 281-290)

  1. Lane 208: change “while” with another term.

-->Thank you for pointing this out. We edited it (Line 284).

  1. Lanes 222-223: do you have any data about CLEIA and ELISA sensitivity among immunocompromised patients? This would be interesting to investigate and include in the manuscript, if possible.

-->Thank you for pointing this out. We agree that it is an interesting if we include immunocompromised patients. We did not have such patients and data at this current study.  We will investigate in the next study if we have such patients. 

Round 2

Reviewer 1 Report

The authors have address all my major comments and the manuscript is much improved. However, I found 24 grammar errors, typographical errors, and formatting errors that should be corrected.

Line 23, change "and enzyme" to "and an enzyme" [grammar]
Lines 25-26, change "reached at plateau" to "reached a plateau" [typo]
Line 26, change "with 3.8" to "with a 3.8" change "and 22-fold" to "and a 222-fold" [grammar]
Line 30, change "especially in low" to "especially in the low" [grammar]
lines 31-32, ">10 BAU/ml" should be placed in parentheses immediately after "high titer range" rather than at the end of the sentence [grammar]
line 62, change "Recent study" to "A recent study" [grammar]
Line 70, change "Other study" to "Another study" [grammar]
Line 90, change "and enzyme-linked" to "and an enzyme-linked"
Line 113, the new paragraph indentation is inconsistent with all other paragraphs
Line 123, change "was reacted with" to "was placed in contact with" [grammar and word choice]
Line 145, spell out the acronyms BSA and PBST
Line 149, spell out the acronym HRP
Line 171, change "were observed as negative control" to "was observed as a negative control" [grammar]
Line 214, change "log-transformed value" to "log-transformed values" [grammar]
Lines 214-215, the blank line between these lines should be removed. [formatting]
Pages 5 and 7, the captions of Figures 1 and 2 have a smaller font size than the caption of Figure 3. I am unsure which font size is correct, but they cannot both be correct.
Line 254, change "supplement" to "Supplement"
Line 295, change "sample size, which potentially" to "sample size potentially limits"
Line 302  change "that the highly" to "that a highly" [grammar]
The blank line between lines 324 and 325 should be removed [formatting]
Reference 9 is missing the page numbers [formatting]
Reference 15 is missing the volume and page numbers [formatting]
Reference 17 is missing the article number [formatting]
Reference 19 should have the journal start with upper case letters and the page numbers are missing [formatting]

Author Response

Dear Prof. Dr. Tripp,

Editor-in Chief

We thank the reviewers for their generous comments on the manuscript and have edited the manuscript to address the concerns. Please see below, in blue, for a point-by-point response to the reviewers’ suggestion. All page numbers/line numbers refer to the revised manuscript file

Line 23, change "and enzyme" to "and an enzyme" [grammar]

-->Thank you for suggestion. We edited it in the manuscript (Line 23).

Lines 25-26, change "reached at plateau" to "reached a plateau" [typo]

--> Thank you for suggestion. We edited it in the manuscript (Line 26).

Line 26, change "with 3.8" to "with a 3.8" change "and 22-fold" to "and a 222-fold" [grammar]

--> Thank you for suggestion. We edited it in the manuscript (Lines 26, 27).

Line 30, change "especially in low" to "especially in the low" [grammar]

--> Thank you for suggestion. We edited it in the manuscript (Line 31).

lines 31-32, ">10 BAU/ml" should be placed in parentheses immediately after "high titer range" rather than at the end of the sentence [grammar]

--> Thank you for suggestion. We edited it in the manuscript (Line 32).

line 62, change "Recent study" to "A recent study" [grammar]

--> Thank you for suggestion. We edited it in the manuscript (Line 63).

Line 70, change "Other study" to "Another study" [grammar]

--> Thank you for suggestion. We edited it in the manuscript (Line 71).

Line 90, change "and enzyme-linked" to "and an enzyme-linked"

--> Thank you for suggestion. We edited it in the manuscript (Line 91).

Line 113, the new paragraph indentation is inconsistent with all other paragraphs

--> Thank you for suggestion. We edited it in the manuscript (Line 116).

Line 123, change "was reacted with" to "was placed in contact with" [grammar and word choice]

--> Thank you for suggestion. We edited it in the manuscript (Line 126).

Line 145, spell out the acronyms BSA and PBST

--> Thank you for suggestion. We included it in the manuscript (Lines 148-149).

Line 149, spell out the acronym HRP

--> Thank you for suggestion. We included it in the manuscript (Line153).

Line 171, change "were observed as negative control" to "was observed as a negative control" [grammar]

--> Thank you for suggestion. We edited it in the manuscript (Line 178).

Line 214, change "log-transformed value" to "log-transformed values" [grammar]

--> Thank you for suggestion. We edited it in the manuscript (Line 225).

Lines 214-215, the blank line between these lines should be removed. [formatting]

--> Thank you for suggestion. We formatted and removed the line.

Pages 5 and 7, the captions of Figures 1 and 2 have a smaller font size than the caption of Figure 3. I am unsure which font size is correct, but they cannot both be correct.

--> Thank you for suggestion. We used same font size in all figures.

Line 254, change "supplement" to "Supplement"

--> Thank you for suggestion. We edited it in the manuscript (Line 264).

Line 295, change "sample size, which potentially" to "sample size potentially limits"

--> Thank you for suggestion. We edited it in the manuscript (Line 306).

Line 302  change "that the highly" to "that a highly" [grammar]

--> Thank you for suggestion. We edited it in the manuscript (Line 313).

The blank line between lines 324 and 325 should be removed [formatting]

--> Thank you for suggestion. We formatted and removed the line (Line 342).

Reference 9 is missing the page numbers [formatting]

--> Thank you for suggestion. We formatted reference No. 9.

Reference 15 is missing the volume and page numbers [formatting]

--> Thank you for suggestion. We formatted reference No. 15.

Reference 17 is missing the article number [formatting]

--> Thank you for suggestion. We formatted reference No. 17.

Reference 19 should have the journal start with upper case letters and the page numbers are missing [formatting]

--> Thank you for suggestion. We formatted reference No. 19.

Sincerely,

Said Ashenagar and Akiko Matsumoto

Reviewer 2 Report

I appreciate that the authors responded to my critiques and addressed the critical points. The manuscript was significantly improved with respect to the previous version. However, I would suggest to make a few corrections as listed below:

  1. It’s not clear what the negative control is represented by. Please, better address this point throughout the manuscript.
  2. Lanes 170-173: it’s not clear, please reframe and include in the material and methods section, "2.1.1. Study design and participants" section, the negative control group.
  3. Was (were) the negative control(s) tested with all the described methodologies? It’s better to stress this point to fully support your data.

Author Response

Dear Prof. Dr. Tripp,

Editor-in Chief

We thank the reviewers for their generous comments on the manuscript and have edited the manuscript to address the concerns.

Lanes 170-173: it’s not clear, please reframe and include in the material and methods section, "2.1.1. Study design and participants" section, the negative control group.

Was (were) the negative control(s) tested with all the described methodologies? It’s better to stress this point to fully support your data.

-->Thank you for the comment. We included the information about negative control in Section 2.1.1 (Lines 105-107).

Sincerely,

Said Ashenagar and Akiko Matsumoto
